# Palladium-catalyzed regio- and enantioselective migratory allylic C(sp$^3$)-H functionalization

Ye-Wei Chen [1,2,3], Yang Liu[1,3], Han-Yu Lu[1,2], Guo-Qiang Lin [1,2✉] & Zhi-Tao He [1✉]

Transition metal-catalyzed asymmetric allylic substitution with a suitably pre-stored leaving group in the substrate is widely used in organic synthesis. In contrast, the enantioselective allylic C(sp$^3$)-H functionalization is more straightforward but far less explored. Here we report a catalytic protocol for the long-standing challenging enantioselective allylic C(sp$^3$)-H functionalization. Through palladium hydride-catalyzed chain-walking and allylic substitution, allylic C-H functionalization of a wide range of acyclic nonconjugated dienes is achieved in high yields (up to 93% yield), high enantioselectivities (up to 98:2 er), and with 100% atom efficiency. Exploring the reactivity of substrates with varying p$K_a$ values uncovers a reasonable scope of nucleophiles and potential factors controlling the reaction. A set of efficient downstream transformations to enantiopure skeletons showcase the practical value of the methodology. Mechanistic experiments corroborate the PdH-catalyzed asymmetric migratory allylic substitution process.

[1] CAS Key Laboratory of Synthetic Chemistry of Natural Substances, Center for Excellence in Molecular Synthesis, Shanghai Institute of Organic Chemistry, University of Chinese Academy of Sciences, Shanghai, China. [2] School of Physical Science and Technology, ShanghaiTech University, Shanghai, China. [3]These authors contributed equally: Ye-Wei Chen, Yang Liu. ✉email: lingq@sioc.ac.cn; hezt@sioc.ac.cn

Transition metal-catalyzed enantioselective allylic substitution has emerged as one of the most useful and reliable transformations for precise stereo- and regiocontrol over a target compound in organic synthesis[1–6]. The constructed carbon−carbon or carbon-heteroatom stereocenter vicinal to a C = C bond can enrich the downstream derivatizations. Diverse transition metals like Pd, Ir, Rh, Ru, Cu, Ni et al have witnessed extensive studies and applications in asymmetric $\eta^3$-allylic substitution[1–6]. In general, an alkene prefunctionalized with an allylic leaving group undergoes oxidative addition to a transition metal to furnish a thermodynamically stable $\eta^3$-$\pi$-allyl metal species. Subsequent outer-sphere nucleophilic attack provides the allylation product stereo- and regioselectively. Alternative precursors, such as allenes, alkynes, and 1,3-dienes, for stereoselective allylations have also achieved considerable progress[7,8].

In contrast, the corresponding allylic C−H functionalization ought to be more straightforward and economic but is far less developed[9,10]. In general, the key $\eta^3$-$\pi$-allyl metal intermediate is generated through direct allylic C−H cleavage (Fig. 1a)[10]. Elegant studies from Rainey[11], Trost[12,13], Gong[14–19] and White[20,21] groups et al. have realized the C−H functionalization of the allylic carbon center that usually requires further activation by another vicinal sp[2] carbon or heteroatom unit. A few studies have been performed on unactivated allylic C−H bond but generally with moderate enantioselectivities[22–25]. Therefore, efficient strategies are still highly desired for enantioselective allylic C−H functionalization, especially for functionalization with inert allylic C−H bonds.

Metal walking[26–35] along an aliphatic chain via iterative $\beta$-hydride elimination and alkene hydrometallation has proven to be very effective in realizing activation at a remote C−H bond[36–41]. Recently, Wu[42] and Zhou[43] developed enantioselective tandem Heck/Tsuji−Trost reaction of 1,4-cyclohexadienes to prepare difunctionalized cyclohexenes[42–49]. In 2020, Fang et al. reported a Ni-catalyzed asymmetric hydrocyanation of skipped dienes, providing allyl nitriles via inner-sphere reductive elimination[33]. Inspired by these elegant studies, we envisioned that with a remote olefin unit prestored in a target allyl compound, the combination of metal hydride-catalyzed[50,51] alkene migration to indirectly realize the allylic C−H activation followed by $\eta^3$-allylation might provide a conceptually different strategy to the challenging allylic C−H functionalization[42–49]. Specifically, a terminal olefin first inserted into a palladium hydride catalyst[52–66] generated facilely in situ and was then transferred to a remote target allylic carbon center via metal walking. The subsequent outer-sphere nucleophilic attack of the resulting $\eta^3$-allyl palladium species would yield an allylic C−H functionalization product (Fig. 1b). However, this 100% atom-economic C−H allylation process is not straightforward. The merger of stereoselective PdH-catalyzed olefin migration and $\eta^3$-allylation is unknown, presumably due to issues from the compatibility of two different processes and the stereoselectivity control inside, especially along a flexible acyclic olefin chain.

Here, we show that the merger of PdH-catalyzed chain-walking and $\eta^3$-allylation serves as an efficient route to the stereoselective activation of the inert allylic C(sp[3])−H bond. A series of flexible acyclic nonconjugated dienes undergo the C(sp[3])−H allylation in high yields and enantioselectivities (up to 93% yield, generally 90−96% ee) and 100% atom-efficiency. Mechanistic studies further provide evidence for the designed migratory allylation strategy.

## Results

**Investigation of reaction conditions**. We initiated the migratory allylic functionalization by studying reactions with an acyclic non-conjugated diene **1a** as the pre-electrophile, diethyl malonate **2a** as the pronucleophile, and Et$_3$N as the base with a palladium catalyst (Fig. 2). A group of chiral ligands were first evaluated for their potential to facilitate both the chain walking and allylic substitution. Trost ligand **L1** which was effective in Pd-catalyzed allylation, failed to furnish the target product **3a** (entry 1)[67]. Trace amounts of **3a** were observed with bisphosphine ligand **L2** in low enantioselectivity, and **L3** did not provide any product (entries 2−3). Because oxazoline-type ligands have been widely adopted in Pd-catalyzed chain-walking reactions[36–41], we turned to **L4** and **L5** for the possibility of migratory allylic substitution. However, none of them led to the desired allylation product **3a** (entries 4−5). Then a set of JosiPhos-type chiral ligands were further evaluated due to their previous application in Pd-catalyzed asymmetric hydrofunctionalization of conjugated dienes (entries 6−9)[58,59]. Fortunately, around 20% yield of **3a** was observed for all cases, and the enantioselectivities were achieved up to 97:3. The reaction yield was finally raised to 82% when Et$_3$N was used as both the base and solvent, though many other solvents also benefited the yield (entries 10−13). Thus, the optimal condition for migratory allylic functionalization was identified as the combination of the skipped diene **1a** (1.1 equiv), nucleophile **2a** (1.0 equiv), [Pd(allyl)Cl]$_2$ (2.5 mol%), **L9** (5 mol%) and NaBArF$_4$ (5 mol%) in Et$_3$N at 60 °C for 24 h.

**Substrate scope**. The scope of substituted nonconjugated diene electrophiles was first checked and the results are summarized in Fig. 3. Skipped dienes with interval methylene chains of different lengths had no discernible influence on the reaction yield and enantioselectivity (**3a–3e**). For example, flexible 1,9-diene as the electrophile delivered C(sp[3])−H allylation product **3e** in 62% yield and 98:2 er. Aryl units with diverse functional groups, such as alkyl, phenyloxy, strained cyclopropyl, silyl, Cl, CF$_3$, CF$_3$O, and OTBDPS, among others, in nonconjugated diene substrates underwent C(sp[3])−H functionalization in 41−91% yield and 91:9−97:3 er (**3f–3m, 3p–3u, 4a**). Aryl-substituted dienes containing strong electron-withdrawing groups, such as sulfonyl or sulfamoyl units, provided the products in moderate yields but

### a. Classical model for $\eta^3$–allylation via C-H activation

### b. Our strategy: migratory allylic substitution

**Fig. 1 Transition metal-catalyzed allylic C−H substitution. a** Classical model for enantioselective transition metal-catalyzed allylic C−H substitution. **b** Our strategy: metal migration triggers allylic C(sp[3])-H functionalization. NuH, nucleophile.

**Fig. 2 Evaluation of reaction conditions for the migratory allylic substitution.** [a]The reactions were run with **2a** (0.2 mmol), NaBAr$^F_4$ (Ar$^F$ = 3,5-(CF$_3$)$_2$Ph, 5 mol%) in solvent (0.2 mL). Yields were determined by crude [1]H NMR. Er values were determined by chiral HPLC. THF, tetrahydrofuran. CPME, cyclopentyl methyl ether. [b]Isolated yield.

| Entry[a] | Ligand | solvent | Yield (%) | er |
|---|---|---|---|---|
| 1 | L1 | Toluene | 0 | |
| 2 | L2 | Toluene | 8 | 62:38 |
| 3 | L3 | Toluene | 0 | |
| 4 | L4 | Toluene | 0 | |
| 5 | L5 | Toluene | 0 | |
| 6 | L6 | Toluene | 23 | 95:5 |
| 7 | L7 | Toluene | 21 | 90:10 |
| 8 | L8 | Toluene | 21 | 97:3 |
| 9 | L9 | Toluene | 29 | 97:3 |
| 10 | L9 | THF | < 5 | 89:11 |
| 11 | L9 | CPME | 34 | 97:3 |
| 12 | L9 | n-Hexane | 52 | 97:3 |
| 13 | L9 | Et$_3$N | 91 (82)[b] | 97:3 |

with high enantioselectivities (**3n**, **3o**). The heteroaryl-tethered acyclic diene **1v** was also suitable for the C(sp$^3$)−H functionalization reaction, producing product **3v** in 93% yield and 96:4 er.

Diverse dicarbonyl derivatives were subsequently tested for migratory allylic functionalization (**3w**−**3z**). Dimethyl malonate **2b** coupled smoothly with different nonconjugated dienes in good yields and high enantioselectivities without regioselectivity issue[60] (**3w**, **3x**). α-Ketoester **2c** was found to be an effective nucleophile, delivering **3y** in 68% yield and 97:3 er, albeit with low diastereoselectivity because of the facile epimerization of the α-carbon center of the dicarbonyl unit. 1,3-Dione **2d** with a lower p$K_a$ showed moderate reactivity and yielded allylation product **3z** in 84:16 er, suggesting that the migratory allylation might be suitable for nucleophiles with a variety of p$K_a$[57].

The utility and robustness of the present protocol were also demonstrated by conducting C(sp$^3$)−H functionalization with a series of dienes derived from complex molecules. Dienes tethered to drugs, such as naproxen, oxaprozin, isoxepac, and febuxostat, efficiently underwent the migratory allylation process to furnish the corresponding products in 57−90% yield with 97:3−98:2 er (**4e**, **4f**, **4i**, **4j**). Electrophiles bearing a Mosher ester, bridged- or

macrocycles, camphanic acid, or ribofuranose derivatives reacted in moderate to good yields with high enantioselectivities (**4b**−**4d**, **4g**−**4h**). Notably, all the aforementioned cases underwent migratory allylation with exclusive regioselectivity at the target allylic carbon center. However, tertiary nucleophiles and a couple of other substrates turned out to be ineffective for this transformation (see Supplementary Methods for details).

To uncover the p$K_a$ range of nucleophiles and the factors controlling reactivity, a series of nucleophiles with p$K_a$ ranging from 9 to 24 were evaluated for migratory functionalization reactions (Fig. 4a). The secondary carbon nucleophiles were chosen and evaluated in order to exclude the steric hindrance as a potential interference factor. As the present protocol contained two processes, i.e., chain walking and allylic substitution, alkene **5**, which could only undergo chain walking, was adopted to test the walking potential of a chosen nucleophile under standard conditions. Using a nucleophile (**2e**−**2h**) with a p$K_a$ < 13 resulted in olefin **5** smoothly undergoing chain-walking to produce **6** in 46−78% yield, but little allylation was observed. By contrast, gradually increasing the p$K_a$ of the nucleophiles from 18 to 24 (**2i**−**2k**), resulted in almost no chain-walking or allylation. Therefore, nucleophiles with p$K_a$ between 13 and 18 appear to be suitable for target migratory allylation (**2a**−**2d**). Our data suggest that the migratory allylation for nucleophiles with p$K_a$ < 13 might be limited by insufficient nucleophilicity, whereas that for nucleophiles with p$K_a$ > 18 does not lead to chain-walking because of insufficient acidity to generate the palladium hydride catalyst.

The mixtures of stereoisomers and regioisomers of skipped dienes underwent regioconvergent migratory allylation in moderate yield and high enantioselectivity (Fig. 4b). To uncover the factor leading to the decreased reactivity, diene **7** or **8** was used as substrate individually under standard condition. The conjugated diene **8** underwent hydrofunctionalization smoothly, giving **3a** in good yield and high er, though 27% of **8** remained unreacted. In contrast, the internal skipped diene **7** showed very low reactivity, providing **3a** in only 4% yield but with high er. We proposed that the coordination of palladium catalyst with the diene substrate might be a crucial factor controlling the results. It should be easier for palladium catalyst to initiate the reaction by coordinating with diene substrate **1a** bearing a terminal olefin than **7** and **8** with sterically bulky internal olefin units. The higher reactivity of **8** than that of **7** was possibly resulted from the quick consumption of the stable π-allyl-Pd intermediate from the migratory insertion of a conjugated diene with Pd−H catalyst.

**Transformations**. The synthetic value of the migratory allylation was highlighted by a gram-scale test and a series of downstream transformations (Fig. 4c). A total of 2.2 gram of compound **3w** were obtained in high yield and enantioselectivity under standard conditions. This compound was then converted to a set of important skeletons in short steps. A combination of decarboxylation and oxidation was used to efficiently prepare γ-lactones **11** and **12** with multiple stereogenic centers, as demonstrated by X-ray crystal analysis. Enantiopure cyclopentene **14** was easily constructed through a metathesis reaction. Moreover, Au(I)-catalyzed [2 + 2] cycloaddition[68] enabled the enantioselective construction of complex bicyclo-[3.2.0] motif **16**, which exists widely in biologically active molecules[69].

**Mechanistic studies**. A set of mechanistic studies were conducted to further elucidate the mechanism of the present protocol (Fig. 5). Quenching the model reaction after only 3 h produced compounds **7** and **8** via alkene migration in 6% and 13% yields, respectively, along with a 25% yield of product **3a** (Fig. 5a). This

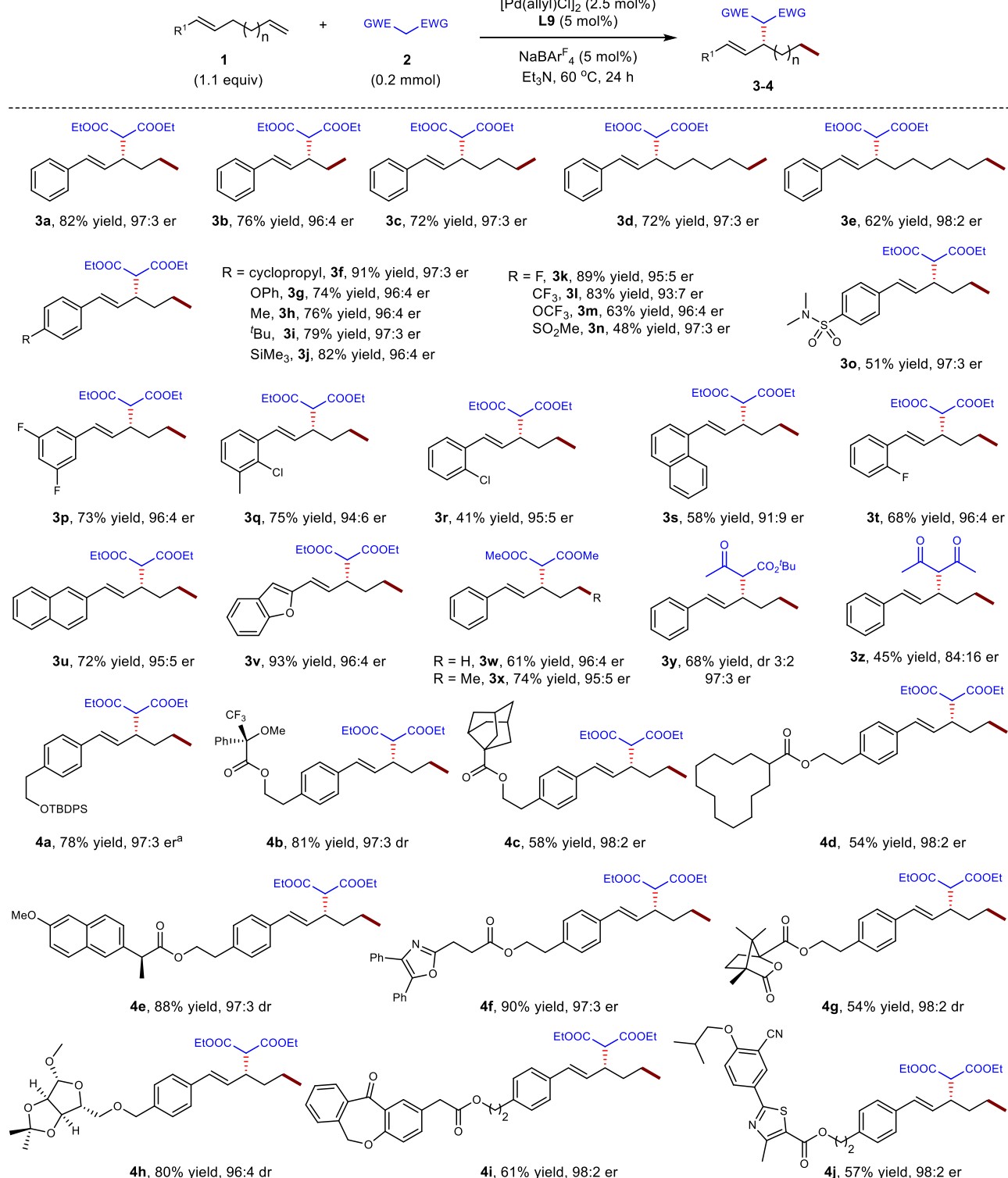

**Fig. 3 Scope of the substrates that undergo migratory allylic substitution.** The reactions were run with **1** (0.22 mmol), **2** (0.2 mmol), [Pd(allyl)Cl]₂ (2.5 mol%), **L9** (5 mol%), NaBAr$^F_4$ (5 mol%) in Et₃N (0.2 mL) at 60 ℃ for 24 h. EWG, electron-withdrawing group. ᵃThe er value was determined by the HPLC analysis of the desilyl derivative of **4a**.

fact supports that chain-walking is involved in the reaction, which is further corroborated by the results of deuteration studies showing D labeling of product *d*-**3a** at multiple carbon centers (Fig. 5b). Moreover, a crossover experiment using deuterated *d*-**1a** and non-deuterated diene **9** as competing electrophiles led to the detection of *d*-**4e** (Fig. 5c). The transfer of D atoms from *d*-**1a** to *d*-**4e** suggests that the PdH catalyst might dissociate from the

ligated diene substrate throughout the chain walking process. Considering the very trace deuteration observed in *d*-**4e**, we deduced that PdH dissociation from the diene substrate might be a challenging and negligible process[38].

Kinetic isotope effect (KIE) was determined to be 1.04, implying that the PdH formation or chain walking process involving hydrogen bond formation or breaking might not be the

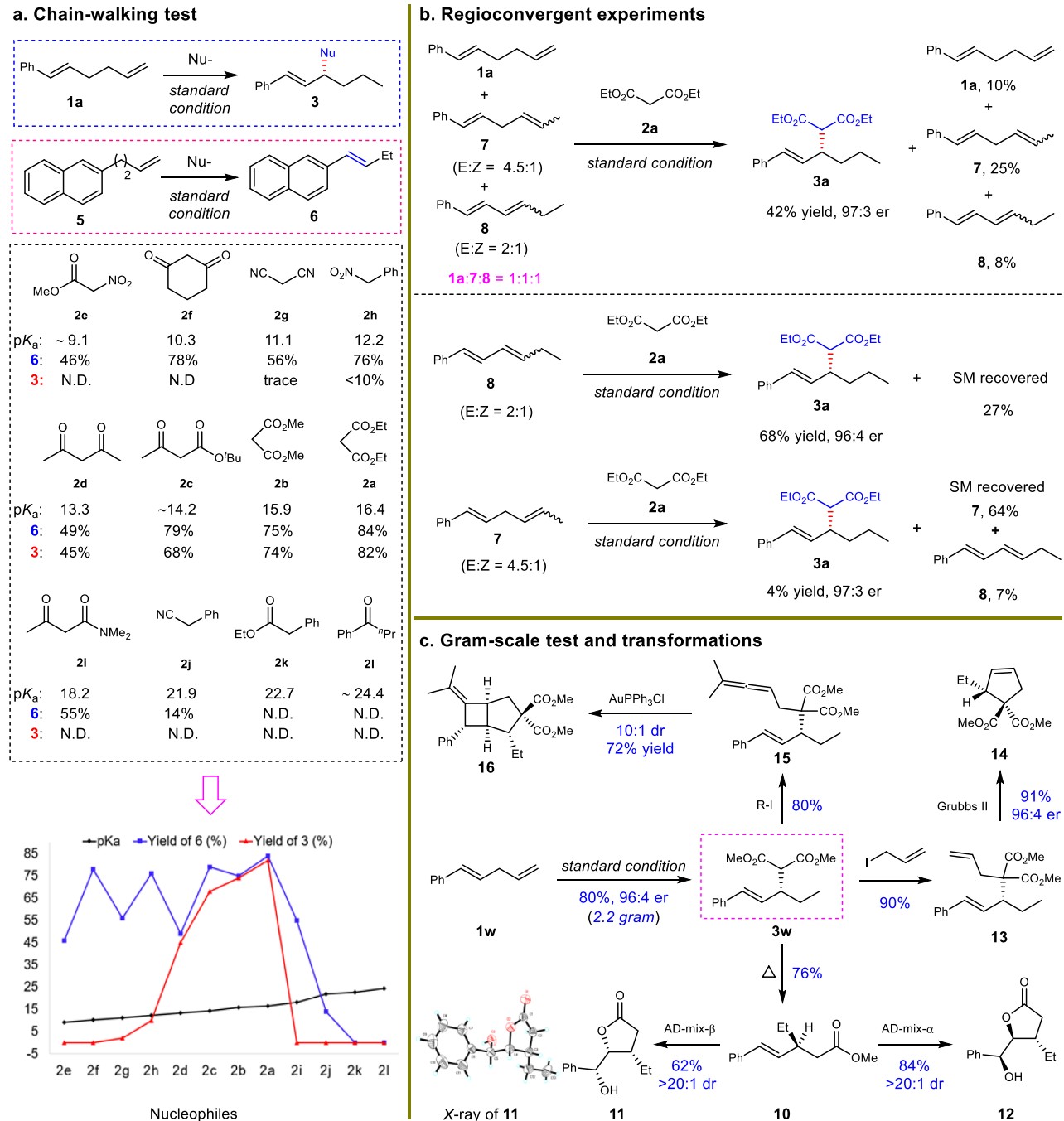

**Fig. 4 Scope of nucleophiles and applications. a** Chain-walking test of nucleophiles. The blue color of the curve represents the yield of compound **6**. The red color of the curve represents the yield of compound **3**. The black color of the curve represents the pKa value of corresponding nucleophiles used. pKa values in DMSO were shown. N.D. not detected. **b** Regioconvergent experiments. **c** Gram-scale test and synthetic application.

rate-determining step (Fig. 5d). Further kinetic studies uncovered that the migratory allylation reaction was first order in both [**cat**] and [**2a**] (Fig. 5e). However, the kinetic data of diene **1a** did not suggest a simple first or zero reaction order, but a saturation kinetics, fitting Michaelis−Menten dynamic model which is commonly used to describe enzymatic reaction[70]. This observation was further supported by a linear relationship between rate$^{-1}$ and [**1a**]$^{-1}$ (Fig. 5e). In order to corroborate the aforementioned facts, we hypothesized the final allylic substitution was the rate-determining step, and deduced the corresponding initial rate equation (see Supplementary Methods for details). This rate law

was consistent with all of the observed kinetic data. Thus, the rate-determining step in the catalytic cycle was determined as the allylic substitution.

In the kinetic studies, an obvious induction period was detected. To elucidate the origin of the induction period, compound **17** from the reaction of nucleophile **2a** with [Pd(allyl)Cl]$_2$ was identified (Fig. 5f). The corresponding initial reaction profile of **17** and product **3a** clearly showed that the yield of **17** increased along with the slow generation of **3a** at the beginning of the reaction. When **17** reached close to the highest 4−5% yield, the reaction rate for **3a** arrived at an obviously higher

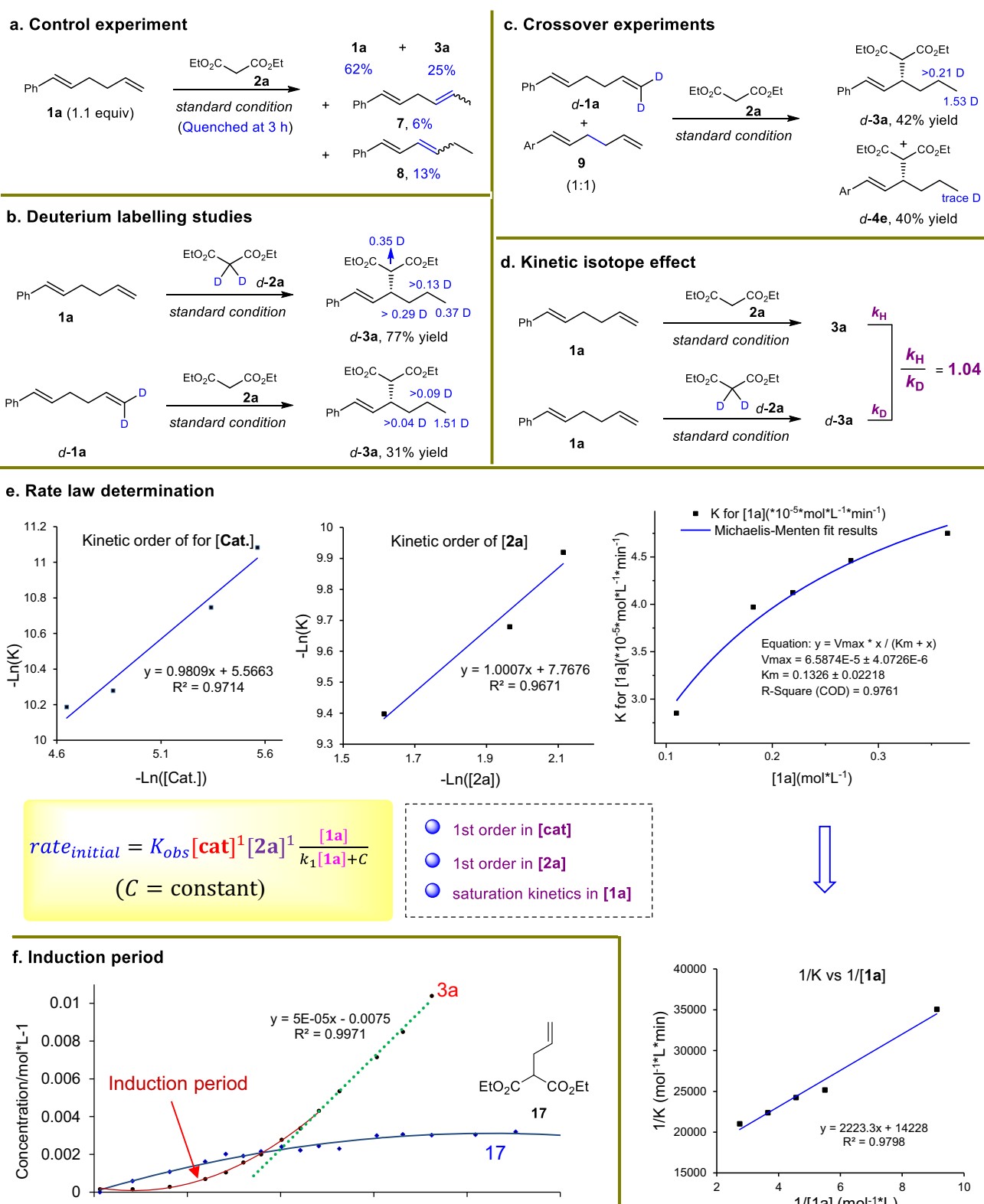

**Fig. 5 Mechanistic studies. a** Control experiments to uncover the reaction intermediates. **b** Deuterium labeling studies show the involvement of olefin migration. **c** Crossover experiments. **d** Kinetic isotope effect. **e** Rate law determination to elucidate the rate-determining step. **f** Initial kinetic studies of **3a** and **17** for induction period determination.

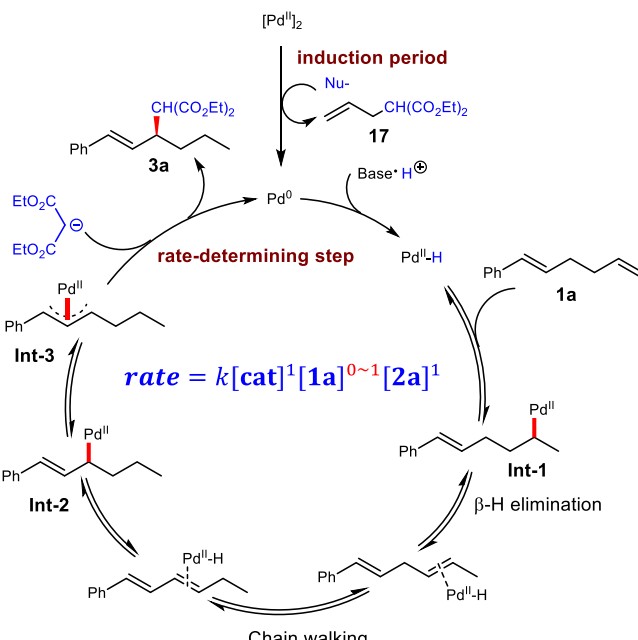

**Fig. 6 Proposed mechanism.** Pd-catalyzed migratory allylic substitution with outer-sphere nucleophiles.

level and maintained stable. Therefore, the induction period derived from the slow formation of active Pd(0) catalyst via off-cycle allylation.

**Proposed mechanism**. Based on the above facts and reported work on the studies of chain walking[36–41], a plausible mechanism was proposed as shown in Fig. 6. The proton would first oxidatively add to Pd(0) to generate PdH species in situ which then inserted into the less sterically hindered alkene to give an unstable alkylpalladium intermediate **Int-1**. After iterative $\beta$-H elimination and hydrometallation, a thermodynamically stable $\eta^3$-$\pi$-allyl Pd species **Int-3** was formed. Finally, the regio- and enantioselective outer-sphere nucleophilic substitution was conducted to deliver the allylation product **3a** and regenerate the Pd(0) catalyst.

## Discussion

In conclusion, we have established a reliable protocol to realize challenging enantioselective acyclic allylic C(sp³)−H functionalization. Hydropalladation is used to trigger stereoselective chain-walking, which is combined with $\eta^3$-allylation to construct a carbon−carbon stereogenic center from an inert C−H bond in high yields and enantioselectivities and 100% atom-efficiency. Studies on the correlation between the p$K_a$ of the nucleophiles with the chain-walking and allylation processes showed the origins of the p$K_a$ range of nucleophiles (p$K_a$ = ~13 to ~18) and the potential factors controlling the reactivity. Importantly, the products could be derivatized to access diverse valuable enantiopure structures of biologically active compounds. Mechanistic studies showcased the designed merger of chain walking and allylic substitution. This strategy might open new avenues for achieving diverse inert C(sp³)−H functionalizations.

## Methods

**General procedure for the Pd-catalyzed migratory allylic functionalization**. To a 4 mL vial in the glovebox under nitrogen were added [Pd(allyl)Cl]₂ (1.8 mg, 0.0050 mmol), **L9** (5.6 mg, 0.010 mmol), sodium tetrakis[3,5-bis(trifluoromethyl) phenyl]borate (NaBArF₄, 8.8 mg, 0.010 mmol) and dry Et₃N (0.2 mL). The mixture was stirred at room temperature for 5 min. Then the skipped diene **1** (0.22 mmol) was added to the solution and the reaction continued to stir for 1 min. Finally, the nucleophile **2** (0.20 mmol) was added to the reaction and the resulting mixture was

stirred at 60 °C for 24 h. After this time, the crude mixture was cooled to room temperature, condensed, and crude ¹H NMR was obtained with dibromomethane (7 μL, 0.1 mmol) as an internal standard to help determine the regioselectivity and conversion. The reaction was further purified by flash column chromatography to afford the pure allylation product **3−4**.

## Data availability

For experimental details and procedures, spectra for all unknown compounds, see supplementary files. The X-ray crystallographic data for **11** (CCDC 2076792), have been deposited at the Cambridge Crystallographic Data Center. These data can be obtained free of charge from The Cambridge Crystallographic Data Center via www.ccdc.cam.ac.uk/data_request/cif.

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

## Acknowledgements

This work was supported by National Natural Science Foundation of China (NSFC 22071262, 21871284, 91956113), Shanghai Rising-Star program (20QA1411300), the Science and Technology Commission of Shanghai Municipality (18401933502), the Strategic Priority Research Program of the Chinese Academy of Sciences (XDB 20020100), CAS Key Laboratory of Synthetic Chemistry of Natural Substances, and Shanghai Institute of Organic Chemistry. We thank Prof. John F. Hartwig (UC Berkeley), Qilong Shen (SIOC), and Zheng Huang (SIOC) for insightful discussions.

## Author contributions

Z.-T.H. conceived the project. Y.-W.C., Y.L., and H.-Y.L. performed the experiments, collected and analyzed the data. G.-Q.L. and Z.-T.H. directed this work. Z.-T.H. wrote the paper with the feedback from all authors.

## Competing interests

The authors declare no competing interests.
