## [Peer Review File · Nature Communications]

Palladium-Catalyzed Regio- and Enantioselective Migratory Allylic C(sp³)-H FunctionalizationReviewers' Comments:

Reviewer #1:

Remarks to the Author:

This work by Lin, He and coworkers is focused on remote hydrofunctionalization of skipped dienes. Although palladium hydride-catalyzed asymmetric reactions have been widely studied, especially in Tsuji-Trost type allylation via the formation of π -allyl Pd species, the combination of this process with metal walking is not known. The work demonstrates the potential of above pathway. Using the chain-walking and allylation cascade, the reaction represents a mechanistically interesting route to realize the asymmetric functionalization of allylic C(sp³)-H bond, which is a very challenging but important area. The scope appears to be broad and the stereoselectivities are high. The authors also uncover the possible correlation between the C-H acidity of nucleophiles and reaction processes, which gives some knowledge on choosing suitable nucleophiles. They can also transform the products to a group of attractive building blocks bearing diverse stereogenic centers. Moreover, they conduct some mechanistic studies to corroborate the claimed migratory allylic substitution. This strategy is very clever and should find broader applications. I believe this work warrants publication in Nature Communications and strongly recommend publication with minor changes.

1. Based on the mechanism, the conjugated diene should be the intermediate in the transformation. Is the conjugated diene effective for allylation under standard condition? This information should help strengthen the design and comments are necessary in the text.
2. The studies on the pKa of different nucleophiles all choose secondary carbon nucleophiles. Is it used to exclude the influence of steric hindrance? If it is right, some clarification is necessary to avoid confusion. Also, how about tertiary carbon nucleophiles?
3. Did the authors try other type of nucleophiles like amines?

Reviewer #2:

Remarks to the Author:

The manuscript by Lin and He describes an enantioselective allylic C(sp³)-H functionalization by means of integrating chain-walking and asymmetric allylic substitution enabled by a chiral palladium complex of JosiPhos-type ligand. A wide scope of acyclic nonconjugated dienes can participate in the reaction, and particularly noteworthy is the merger of a stereoselective PdH-mediated olefin migration and η^3 -allylation, enabling a 100% atom economic C-H allylation process. Moreover, the authors have also found the apparent impact of the pKa of nucleophiles on the reaction performance. This reviewer reads the manuscript with great interest and believes that the current manuscript demonstrates significant advances in Pd-catalyzed olefin functionalization. It is a very impressive and beautiful work. Therefore, I recommend this manuscript to be published in Nature Communications after suitable minor revisions by considering following concerns:

1. Given that the nucleophiles with pKa between 13-18 is suitable for this reaction, the results with other type of nucleophiles, such as amides and phenols, are welcome.
2. In addition, it will be more interesting if other substituted dienes, for example, 6-methylhepta-1,5-diene, (E)-hepta-1,5-diene, ((1E,3E)-octa-1,3,7-trien-1-yl)benzene, (E)-octa-3,7-dien-1-yn-1-ylbenzene, (E)-(5-methylhexa-1,5-dien-1-yl)benzene and ((1E,5E)-hepta-1,5-dien-1-yl)benzene, are investigated. If these substrates are unreactive, simple comments should be given in the manuscript, but the results do not weaken the quality of this work.
3. For the mechanistic studies, kinetic experiments are suggested to be conducted for the identification of the rate-limiting step.

Other minor changes:

In Scheme 1a, the terms of "outer-sphere nucleophiles" and "high-valent transition metal" are not appropriate. The palladium catalysis with trivalent phosphorus ligands has been identified to undergo the allylic C-H cleavage via a 16-electron Pd(0) intermediate through a concerted proton and two-electron transfer process (Acc. Chem. Res. 2020, 53, 2841), and in most of cases, the inner-sphere bond-forming pathway prefers to provide the branch-selectivity.

Reviewer #3:

Remarks to the Author:

In this manuscript, He and Lin report a Pd-catalyzed remote enantioselective alkylation of non-conjugated dienes with both high regio- and enantioselectivity, through the combination of alkene isomerization and subsequent nucleophilic attack. Although this strategy is not new and was developed earlier by Larock and co-workers (*J. Org. Chem.* 1991, 56, 15, 4589), related studies of enantioselective functionalizations involving olefin chain walking using acyclic skipped dienes are still limited. The most relevant report to the present study was done by Fang and co-workers (*Angew. Chem. Int. Ed.* 2020, 59, 21436), and only enantioselective hydrocyanation was achieved. This manuscript represents a good contribution to the field of enantioselective functionalizations of non-conjugated dienes. Therefore, the work reported in this manuscript has considerable interest for the readers of *Nat. Commun.* However, a number of issues need to be addressed before the manuscript can be considered for publication:

1. The substrate scope of skipped dienes in Table 2 is commendable, but R1 appears to be limited to aryl units. How efficient and enantioselective are dienes with aliphatic groups (R1 = alkyl)? Also, do skipped dienes enclosed within a ring work too? Any limitations should be commented in the paper.
2. The scope of nucleophiles is somewhat limited to 1,3-dicarbonyl compounds of a certain pKa range. In Table 3, for the nucleophiles that are effective (2b-d), what are the enantioselectivities of the corresponding products 3? Please consider drawing the general structure of 3 in the equation below Table 3 to minimize confusion.
3. Did the authors examine 2b-d with a skipped diene (e.g. 1a)?
4. In Table 3, the yields of 3 and 6 derived from the reaction of 5 with 2a-d appear to be inaccurate. Alkene 5 was presumably employed in 1.1 equiv., yet the total combined yields of 3 and 6 exceeded the theoretical 110%. Please do a thorough check on these results. The same issue exists for the control experiment in Fig. 2a.
5. For those nucleophiles that have $pK_a > 18$, the authors claimed that migratory allylation was inefficient due to the lack of chain-walking as a consequence of insufficient acidity to generate the Pd-H catalyst. Did the authors examine stronger organic bases other than Et₃N?
6. External hydride reagents (e.g. hydrosilanes) have been employed to promote metal-catalyzed alkene isomerizations. For those nucleophiles that are ineffective to promote chain-walking, did the authors attempt to utilize an external hydride reagent to induce isomerization instead prior to allylation?
7. To demonstrate versatility, the authors should consider performing a regioconvergent experiment using a mixture of regioisomeric diene substrates and show that they converge to a single product.
8. For the references section, many formatting errors could be found such as 1e, 7a, 8.... Please ensure that the same format applies throughout all references.
9. The signal from CDCl₃ appears to be stronger than usual in the ²H NMR spectrum of 4e.

Point-by-point responses to the reviewers

Reviewer 1

1. The reviewer wondered if the conjugated diene was effective for the allylation reaction. We tested the conjugated diene as the substrate under standard condition. Good yield and comparable high ee of product **3a** were observed.

The related comments were added to the text as described:

“The conjugated diene **8** underwent hydrofunctionalization smoothly, giving **3a** in good yield and high er, though 27% of **8** remained unconsumed.”

2. The reviewer asked if the secondary carbon nucleophile was used to exclude the influence of steric hindrance. The reviewer is right and the related comments were added to the text as shown below:

“The secondary carbon nucleophiles were chosen and evaluated in order to exclude the steric hindrance as a potential interference factor.”

3. The reviewer asked how about tertiary carbon nucleophiles in the reaction. We did test this type of nucleophiles. Unfortunately, only trace products were observed by crude ¹H NMR and GC-MS. We added this fact to the supplementary information.

Thus, the related comments were added to the text as shown below:

“In addition to the scope of diene substrates and nucleophiles described above, a couple of substrates ineffective for this transformation were also summarized in supplementary information.”

4. The reviewer wondered if other nucleophiles like amine was tested for this migratory allylation reaction. We did check secondary amine like morpholine as a potential nucleophile. Moderate

yield was observed but in low ee. This fact as below was added to the supplementary information.

Reviewer 2

1. The reviewer wondered if other nucleophiles like amides or phenols were reactive. We tested these two types of nucleophiles, but did not observe any product. For the amide substrate, the corresponding pK_a value (>18.2) is too high and the substrate is also bulkier than dimethyl malonate. For the phenol, the according pK_a value (~ 17) might be suitable, yet no allylation product was observed except for 59% alkene isomerization intermediate. The construction of C-O bond is challenging due to the known worse nucleophilicity of oxygen in allylation. This fact as below was added to the supplementary information.

2. The reviewer asked if a group of special substrates were reactive. We prepared these substrates as the reviewer suggested and evaluated the corresponding reactivity as shown below.

All of these results and more types of substrates used were summarized in supplementary information. The related comments were added to the text as shown below:

“In addition to the scope of diene substrates and nucleophiles described above, a couple of substrates ineffective for this transformation were also summarized in supplementary information.”

- The reviewer wondered the rate-limiting step of the reaction. A set of kinetic studies were conducted to elucidate the KIE value, reaction orders and reduction period. All of these data suggested that the allylic substitution in the catalytic cycle might be the rate-determining step. These facts were added to the text and supplementary information. The related comments and figures were added to the text as shown below:

Fig. 3 Mechanistic studies. a Control experiments. b Deuterium labelling studies. c Crossover experiments. d Kinetic isotope effect. e Rate law determination. f Initial kinetic studies of **3a** and **17** for reduction period determination.

“Kinetic isotope effect (KIE) was determined to be 1.04, implying that the PdH formation or chain walking process involving hydrogen bond formation or breaking might not be the rate-determining step (Fig. 3d). Further kinetic studies uncovered that the migratory allylation reaction was first order in both [cat] and [2a] (Fig. 3e). However, the kinetic data of diene **1a** did

not suggest a simple first or zero reaction order, but a saturation kinetics, perfectly fitting Michaelis-Menten dynamic model which is commonly used to describe enzymatic reaction.⁷⁰ This observation was further supported by a linear relationship between rate⁻¹ and [1a]⁻¹ (Fig. 3e). In order to corroborate the aforementioned facts, we hypothesized the final allylic substitution was the rate-determining step, and deduced the corresponding initial rate equation (see SI for details). This rate law was consistent with all of the observed kinetic data. Thus, the rate-limiting step in the catalytic cycle was determined as the allylic substitution.

In the kinetic studies, an obvious reduction period was detected. To elucidate the origin of the reduction period, compound 17 from the reaction of nucleophile 2a with [Pd(allyl)Cl]₂ was identified (Fig. 3f). The corresponding initial reaction profile of 17 and product 3a clearly showed that the yield of 17 increased along with the slow generation of 3a at the beginning of reaction. When 17 reached close to the highest 4-5% yield, the reaction rate for 3a arrived at an obviously higher level and maintained stable. Therefore, the reduction period derived from the slow formation of active Pd(0) catalyst via off-cycle allylation.”

4. The reviewer stated that the terms of “outer-sphere nucleophiles” and “high-valent transition metal” are not appropriate. We revised them in the text and Figure 1 as suggested. Now it reads:

“In general, the key η^3 - π -allyl metal intermediate is generated through direct allylic C-H cleavage (Fig. 1a).^{3b}”

Reviewer 3

1. The reviewer asked if alkyl substituent instead of aryl group in the diene substrate or the skipped diene within a ring were reactive. We checked three related substrates. For 1,4-cyclohexadiene or 1,5-cyclooctadiene, they did not undergo expected migratory allylation. When alkyl substituted diene, such as hexa-1,4-diene was used, a group of regioisomers were observed, due to the low discrimination in alkene migration direction.

All of these results and more types of substrates were summarized in supplementary information. The related comments were added to the text as shown below:

“In addition to the scope of diene substrates and nucleophiles described above, a couple of substrates ineffective for this transformation were also summarized in supplementary information.”

2. The reviewer wondered the enantioselectivities of the corresponding products **3** by using **2b-2d** as nucleophiles in Fig. 2a. Actually, these results were summarized in Table 2 as part of the substrate scope checking (**3x-3z**). In order to avoid the confusion from the reviewer and others, we redraw this table as shown below:

3. The reviewer asked the results of migratory allylation by using nucleophiles **2b-2d**. These results were reported in Table 2 as part of the substrate scope checking (**3x-3z**).
4. The reviewer wondered why the total combined yields of **3** and **6** exceeded the theoretical 110%? In fact, it is not suitable to combine the yields of **3** and **6**, because they are two separate parallel reactions. In order to avoid the confusion brought to this reviewer and others, we redraw Fig. 2a as shown above.

For the control experiment shown in Fig. 2b, the total combined yield is 111%, almost the same as the amount of substrate used (1.1 equiv). we rechecked the original data and re-integrated the peaks. Now the data was revised and shown as below:

5. The reviewer asked if the stronger base than Et_3N was checked for nucleophiles that have $\text{p}K_a > 18$. We tested DBU and NaO^tBu as stronger bases with ethyl 2-phenylacetate (2k) as the nucleophile respectively. No product was observed, probably due to the strong basic condition unfavorable for the generation of PdH catalyst (*J. Am. Chem. Soc.* 2019, 141, 14554; *J. Am. Chem. Soc.* 2021, 143, 7285).

6. The reviewer wondered if external hydride reagents like silanes could be used to help induce alkene isomerization. We used 2k as the nucleophile with a high $\text{p}K_a$ value and triethylsilane as the external hydride reagent, but did not observed the expected product.

On the other side, $^i\text{PrOH}$ is also widely used in promoting the generation palladium hydride. Thus, $^i\text{PrOH}$ (1 equiv) was adopted as additives under standard condition. The alkene isomerization product **6** was obtained in 86% yield, proving that $^i\text{PrOH}$ is indeed efficient in facilitating the migration. However, decreased yield was observed with comparable enantioselectivity in the migratory allylation process. In addition, with 2k as the nucleophile, no allylation product was observed.

7. The reviewer suggested a regioconvergent reaction to show the versatility of the transformation. We conducted the regioconvergent reaction and observed moderate yield and the same 97:3 er for the transformation.

Further studies of each diene substrates elucidated that the internal skipped diene **7** was ineffective for the expected functionalization (see the figure above). The results were added to the text and related comments were shown as below:

“The mixtures of stereoisomers and regioisomers of skipped dienes underwent regioconvergent migratory allylation in moderate yield and high enantioselectivity (Fig. 2b). To uncover the factor leading to the decreased reactivity, diene **7** or **8** was used as substrate individually under standard condition. The conjugated diene **8** underwent hydrofunctionalization smoothly, giving **3a** in good yield and high er, though 27% of **8** remained unreacted. In contrast, the internal skipped diene **7** showed very low reactivity, providing **3a** in only 4% yield but with high er. We proposed that the coordination of palladium catalyst with the diene substrate might be a crucial factor controlling the results. It should be easier for palladium catalyst to initiate the reaction by coordinating with diene substrate **1a** bearing a terminal olefin than **7** and **8** with sterically bulky internal olefin units. The higher reactivity of **8** than that of **7** was possibly resulted from the quick consumption of π -allyl-Pd intermediate from the migratory insertion of conjugated diene **8** with Pd-H catalyst.”

8. The reviewer pointed out the format error in references. We revised all of them as suggested.
9. The reviewer asked why the signal from CDCl₃ appeared to be stronger than usual in the ²H NMR spectrum of **4e**. In fact, we enlarged the spectrum in many times in order to show the trace signal of D atom in **4e**. If the spectrum came back to original normal size, the related trace deuterium signal was difficult to be observed.

Reviewers' Comments:

Reviewer #1:

Remarks to the Author:

In the revised manuscript, the authors have addressed all the reviewers questions properly, particularly provided extensive experimental results to discuss the mechanism. This reviewer agree that the revised manuscript is suitable to publish in Nature Communications. But some revisions still should be done before its publications:

1. In this reviewer's eyes, the revised synthetic application part (Fig.2C) is rather disorder indeed. This reviewer suggest to optimize this part again.
2. The authors describe the title of Fig 4 "PdH-catalyzed..." But I think "Pd-catalyzed..." is better, because the reaction is initiated by a Pd(0) species but not a Pd(II)-H.

Reviewer #2:

Remarks to the Author:

The revised manuscript by He and Lin has addressed most of concerns. In particular, many additional experiments have been complemented to address the substrate scope and mechanism issues. It now merits the publication in Nature communications. However, some minor issues remain to be considered:

- (1) In page 2, left column, in the second paragraph, "with an acyclic non-conjugated diene 1a as the electrophile" is not really accurate. "diene" is not an electrophile, but a pre-electrophile. Please correct it.
- (2) In page 2, right column, the description "In addition to the scope of diene substrates and nucleophiles described above, a couple of substrates ineffective for this transformation were also summarized in supplementary information" is not understandable, and is suggested to be revised as "However, tertiary nucleophiles turned out to be ineffective for this transformation (see supplementary information).

Reviewer #3:

Remarks to the Author:

The authors have fully addressed my concerns in the revised manuscript, and it is in my view that the work can now be supported for publication.

Point-by-point responses to the reviewers

Reviewer 1

1. The reviewer thought the transformation figure in Fig. 2c is disordered. We have revised it to make it clearer.
2. The reviewer suggested to use term “Pd-catalyzed” instead of original “PdH-catalyzed”. We have revised it as suggested

Reviewer 2

1. The reviewer thought the use of “pre-electrophile” was better than “electrophile”. We agreed with the reviewer and thus revised it as suggested.
2. The reviewer suggested to revise a sentence for better understanding. We have revised it as shown below:

“However, tertiary nucleophiles and a couple of other substrates turned out to be ineffective for this transformation (see supplementary information).”